# Data-efficient Deep Reinforcement Learning for Dexterous Manipulation

## Abstract

Grasping an object and precisely stacking it on another is a difficult task for traditional robotic control or hand-engineered approaches. Here we examine the problem in simulation and provide techniques aimed at solving it via deep reinforcement learning. We introduce two straightforward extensions to the Deep Deterministic Policy Gradient algorithm (DDPG), which make it significantly more data-efficient and scalable. Our results show that by making extensive use of off-policy data and replay, it is possible to find high-performance control policies that successfully achieve precise stacking behaviour in $> 95\%$ of 1000 randomly initialized configurations. Further, our results on data efficiency hint that it may soon be feasible to train successful stacking policies by collecting interactions on real robots.

## 1 Introduction

Dexterous manipulation is a fundamental challenge in robotics. Researchers have long sought a way to enable robots to robustly and flexibly interact with fixed and free objects of different shapes, materials, and surface properties in the context of a broad range of tasks and environmental conditions. Such flexibility is very difficult to achieve with manually designed controllers. The recent resurgence of neural networks and "deep learning" has inspired hope that these methods will be as effective in the control domain as they are for perception. Indeed, recent work has used neural networks to learn solutions to a variety of control problems (Lillicrap et al., 2016; Schulman et al., 2016; Gu et al., 2016c; Schulman et al., 2015; Heess et al., 2015; Levine & Abbeel, 2014).

While the flexibility and generality of learning approaches is promising for robotics, these methods typically require a large amount of data that grows with the complexity of the task. What is feasible on a simulated system, where hundreds of millions of control steps are possible (Mnih et al., 2016; Schulman et al., 2016), does not necessarily transfer to real robot applications due to unrealistic learning times. One solution to this problem is to restrict the generality of the controller by incorporating task specific knowledge, e.g. in the form of dynamic movement primitives (Schaal, 2006), or in the form of strong teaching signals, e.g. kinesthetic teaching of trajectories (Muelling et al., 2013). Recent works have had success learning flexible neural network policies directly on real robots (e.g. (Levine et al., 2015; Gu et al., 2016a; Yahya et al., 2016)), but tasks as complex as precise grasping-and-stacking remain daunting.

In this paper we investigate in simulation the possibility of learning precise manipulation skills end-to-end with a general purpose model-free deep reinforcement learning algorithm. We assess the feasibility of performing analogous experiments on real robotics hardware and provide guidance with respect to the choice of learning algorithm, experimental setup, and the performance that we can hope to achieve.

We consider the task of picking up a Lego brick from the table and stacking it onto a second nearby brick using a robotic arm and gripper. This task involves contact-rich interactions between the robotic arm and two freely moving objects. It also requires mastering several sub-skills (reaching, grasping, lifting, and stacking). Each of these sub-skills is challenging in its own right as they require both precision (for instance, successful stacking requires accurate alignment of the two bricks) and as well as robust generalization over a large state space (e.g. different initial positions of the bricks and the initial configuration of the arm). Finally, there exist non-trivial and long-ranging dependencies between the solutions for different sub-tasks: for instance, the ability to successfully stack the brick depends critically on having picked up the brick in a sensible way beforehand.

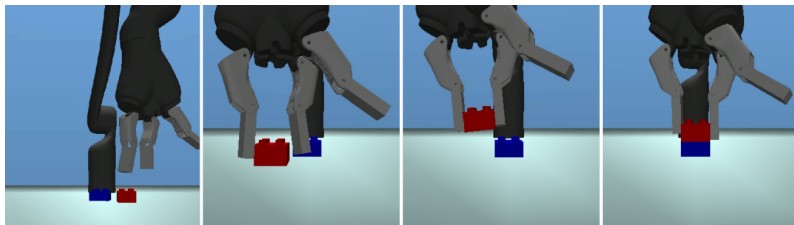

Figure 1: Simulation rendering of the Lego task in different completion stages (also corresponding to different subtasks): (a) starting state, (b) reaching, (c) grasping, and (d) stacking

This paper makes several contributions: 1. We build on the Deep Deterministic Policy Gradient (DDPG; (Lillicrap et al., 2016)), a general purpose model-free reinforcement learning algorithm for continuous actions, and extend it in two ways: firstly, we improve the data efficiency of the algorithm by scheduling updates of the network parameters independently of interactions with the environment. Secondly, we overcome the computational and experimental bottlenecks of single-machine single-robot learning by introducing a distributed version of DDPG which allows data collection and network training to be spread out over multiple computers and robots. 2. We show how to use these straightforward algorithmic developments to solve a complex, multi-stage manipulation problem. We further propose two broadly applicable strategies that allow us to reliably find solutions to complex tasks and further reduce the amount of environmental interaction. The first of these strategies is a recipe for designing effective shaping rewards for compositional tasks, while the second biases the distribution of initial states to achieve an effect akin a form of apprenticeship learning.

In combination these contributions allow us to reliably learn robust policies for the full stacking task from scratch in less than 10 million environment transitions. This corresponds to less than 10 hours of interaction time on 16 robots. In addition, we show that when states from demonstration trajectories are used as the start states for learning trials the full task can be learned with 1 million transitions (i.e. less than 1 hour of interaction on 16 robots). To our knowledge our results provide the first demonstration of end-to-end learning for a complex manipulation problem involving multiple freely moving objects. They are also suggest that it may be possible to learn such non-trivial manipulation skills directly on real robots.

## 2 RELATED WORK

Reinforcement learning (RL) approaches solve tasks through repeated interactions with the environment guided by a reward signal of success or failure (Sutton & Barto, 1998). A distinction is often made between value-based and policy search methods. The latter have been routinely applied in robotics, in part because they straightforwardly handle continuous and high-dimensional action spaces (Deisenroth et al., 2013), and applications include manipulation (Peters & Schaal, 2006; Kalakrishnan et al., 2011; Pastor et al., 2011; van Hoof et al., 2015; Levine et al., 2015; Gu et al., 2016a; Yahya et al., 2016; Gupta et al., 2016), locomotion e.g. (Kohl & Stone, 2004; Matsubara et al., 2006), and a range of other challenges such as helicopter flight (Bagnell & Schneider, 2001). However, policy search methods can scale poorly with the number of parameters that need to be estimated, requiring the need for restricted policy classes, that in turn might not be powerful enough for solving complex tasks.

One exception are guided policy search methods (GPS) (Levine et al., 2015; Yahya et al., 2016). These employ a teacher algorithm to locally optimize trajectories which are then summarized by a neural network policy. They gain data-efficiency by employing aggressive local policy updates and extensive training of their neural network policy. The teacher can use model-based (Levine et al., 2015) or model-free (Yahya et al., 2016) trajectory optimization. The former can struggle with strong discontinuities in the dynamics, and both rely on access to a well defined and fully observed state space.

Alternatively, model-free value function approaches enable effective reuse of data and do not require full access to the state space or to a model of the environment. The use of rich function approximators such as neural networks in value function methods dates back many years, e.g. (Webros, 1990; Tesauro, 1995; Hunt et al., 1992; Hafner & Riedmiller, 2007), and recent success with deep learning has driven the development of new end-to-end training methods for challenging control problems (Mnih et al., 2015; Gu et al., 2016b;c; Lillicrap et al., 2016). Closely related to the ideas followed

in this paper, (Gu et al., 2016a) demonstrates that value-based methods using neural network approximators can be used for relatively simple robotic manipulation tasks in the real world (Gu et al., 2016c). This work also followed a recent trend towards the use of experimental rigs that allow parallelized data collection, e.g. (Pinto & Gupta, 2015), via the use of multiple robots from which experience is gathered simultaneously (Levine et al., 2016; Gu et al., 2016a; Yahya et al., 2016).

Finally, the use of demonstration data has played an important role in robot learning, both as a means to obtain suitable cost functions (Boularias et al., 2011; Kalakrishnan et al., 2013; Finn et al., 2016; Gupta et al., 2016) but also to bootstrap and thus speed up learning. For the latter, kinesthetic teaching is widely used (Peters & Schaal, 2006; Kalakrishnan et al., 2011; Pastor et al., 2011; Yahya et al., 2016), though the need for a human operator to be able to guide the robot through the full movement can be limiting.

## 3 BACKGROUND

In this section we explain the learning problem and summarize the DDPG algorithm. We explain its relationship to other Q-function based RL algorithms in the Appendix.

The RL problem consists of an agent interacting with an environment in a sequential manner to maximize the expected sum of rewards. At time $t$ the agent observes the state $x_t$ of the system and produces a control $u_t = \pi(x_t; \theta)$ according to policy $\pi$ with parameters $\theta$. This leads the environment to transition to a new state $x_{t+1}$ according to the dynamics $x_{t+1} \sim p(\cdot|x_t, u_t)$, and the agent receives a reward $r_t = r(x_t, u_t)$. The goal is to maximize the expected sum of discounted rewards $J(\theta) = \mathbb{E}_{\tau \sim \rho_\theta} \left[ \sum_t \gamma^{t-1} r(x_t, u_t) \right]$, where $\rho_\theta$ is the distribution over trajectories $\tau = (x_0, u_0, x_1, u_1, \dots)$ induced by the current policy: $\rho_\theta(\tau) = p(x_0) \prod_{t>0} p(x_t|x_{t-1}, \pi(x_{t-1}; \theta))$.

DPG (Silver et al., 2014) is a policy gradient algorithm for continuous action spaces that improves the deterministic policy function $\pi$ via backpropagation of the action-value gradient from a learned approximation to the $Q$-function. Specifically, DPG maintains a parametric approximation $Q(x_t, u_t; \phi)$ to the action value function $Q^\pi(x_t, u_t)$ associated with $\pi$ and $\phi$ is chosen to minimize

$$\mathbb{E}_{(x_t, u_t, x_{t+1}) \sim \bar{\rho}} \left[ (Q(x_t, u_t; \phi) - y_t)^2 \right] \tag{1}$$

where $y_t = r(x_t, u_t) + \gamma Q(x_{t+1}, \pi(x_{t+1}))$. $\bar{\rho}$ is usually close to the marginal transition distribution induced by $\pi$ but often not identical. For instance, during learning $u_t$ may be chosen to be a noisy version of $\pi(x_t; \theta)$, e.g. $u_t = \pi(x_t; \theta) + \epsilon$ where $\epsilon \sim \mathcal{N}(0, \sigma^2)$ and $\bar{\rho}$ is then the transition distribution induced by this noisy policy. The policy parameters $\theta$ are then updated according to

$$\Delta\theta \propto \mathbb{E}_{(x,u) \sim \bar{\rho}} \left[ \frac{\partial}{\partial u} Q(x, u; \phi) \frac{\partial}{\partial \theta} \pi(x; \theta) \right]. \tag{2}$$

DDPG (Lillicrap et al., 2016) incorporates experience replay and target networks to the original DPG algorithm: Experience is collected into a buffer and updates to $\theta$ and $\phi$ (eqs. 1, 2) are computed using mini-batch updates with samples from this buffer. A second set of "target-networks" is maintained with parameters $\theta'$ and $\phi'$. These are used to compute $y_t$ in eqn. (1) and their parameters are slowly updated towards the current parameters $\theta, \phi$. Both measures significantly improve the stability of DDPG.

The use of a Q-function facilitates off-policy learning. This decouples the collection of experience data from the updates of the policy and value networks which allows us to make many parameter update steps per step in the environment, ensuring that the networks are well fit to the data that is currently available.

## 4 TASK AND EXPERIMENTAL SETUP

The full task that we consider in this paper is to use the arm to pick up one Lego brick from the table and stack it onto the remaining brick. This "composite" task can be decomposed into several subtasks, including grasping and stacking. We consider the full task as well as the two sub-tasks in isolation:

|  | Starting state | Reward |
|---|---|---|
| Grasp | Both bricks on table | Brick 1 above table |
| StackInHand | Brick 1 in gripper | Bricks stacked |
| Stack | Both bricks on table | Bricks stacked |

In every episode the arm starts in a random configuration with an appropriate positioning of gripper and brick. We implement the experiments in a physically plausible simulation in MuJoCo (Todorov et al., 2012) with the simulated arm being closely matched to a real-world Jaco arm setup in our lab. Episodes are terminated after 150 steps of 50ms of physical simulation time. The agent thus has 7.5 seconds to perform the task. Unless otherwise noted we give a reward of one upon successful completion of the task and zero otherwise.

The observation contains information about the angles and angular velocities of the 6 joints of the arm and 3 fingers of the gripper, as well as the position and orientation of the two bricks and relative distances of the two bricks to the pinch position of the gripper (roughly the position where the fingertips would meet if the fingers are closed). The 9-dimensional continuous action directly sets the velocities of the arm and finger joints. In experiments not reported in this paper we have tried using observations with only the raw state of the brick and the arm configuration (i.e. without the vector between the end-effector and brick) This increased the number of environment interactions needed roughly by a factor of two to three.

For each experimental condition we optimize the learning rate and train and measure the performance of 10 agents with different random initial network parameters. After every 30 training episodes the agent is evaluated for 10 episodes. We used the mean performance at each evaluation phase as the performance measure presented in all plots. In the plots the line shows the mean performance across agents and the shaded regions correspond to the range between the worst and best performing one In all plots the x-axis represents the number of environment transitions seen so far at an evaluation point (in millions) and the y-axis represent episode return.

A video of the full setup and examples of policies solving the component and full tasks can be found here: https://www.youtube.com/watch?v=7vmXOGwLq24.

## 5 ASYNCHRONOUS DPG WITH VARIABLE REPLAY STEPS

In this section we study two methods for extending the DDPG algorithm and find that they can have significant effect on data and computation efficiency, in some cases making the difference between finding a solution to a task or not.

**Multiple mini-batch replay steps** Deep neural networks can require many steps of gradient descent to converge. In a supervised learning setting this affects purely computation time. In reinforcement learning, however, neural network training is interleaved with the acquisition of interaction experience giving rise to a complex interaction. To gain a better understanding of this effect we modified the original DDPG algorithm as described in (Lillicrap et al., 2016) to perform a fixed but configurable number of mini-batch updates per step in the environment. In (Lillicrap et al., 2016) one update was performed after each new interaction step.

We refer to DDPG with a configurable number of update steps as DPG-R and tested the impact of this modification on the two primitive tasks Grasp and StackInHand. The results are shown in Fig. 2 (left). The number of update steps has a dramatic effect on the amount of experience data required. After one million interactions the original version of DDPG with a single update step (blue traces) appears to have made no progress towards a successful policy for stacking, and only a small number of controllers have learned to grasp. Increasing the number of updates per interaction to 5 greatly improves the results (green traces), and with 40 updates (purple) the first successful policies for stacking and grasping are obtained after 200,000 and 300,000 interactions respectively (corresponding to 1,300 and 2,000 episodes). Notably, for both tasks we continue to see a reduction in total environment interaction up to 40 update steps, the maximum used in the experiment.

One possible explanation for this effect is the interaction alluded to above: insufficient training may lead to a form of underfitting of the policy. Since the policy is then used for exploration this affects the quality of the data collected in the next iteration which in turn has an effect on training in future iterations leading to overall slow learning.

We have observed in various experiments (not shown) that other aspects of the network architecture (layer sizes, non-linearities) can similarly affect learning speed. Finally, it is important to note that one cannot replicate the effect of multiple replay steps simply by increasing the learning rate. In practice we find that attempts to do so make training unstable.

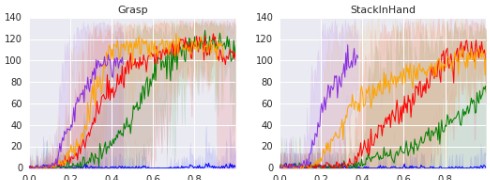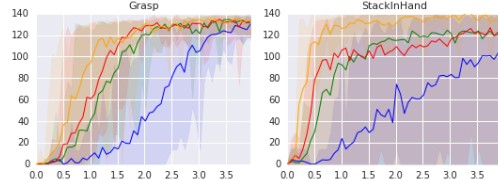

Figure 2: **Left**: (a,b) Mean episode return as a function of number of transitions seen (in millions) of DPG-R (single worker) on the Grasp (left) and StackInHand (right) task with 1 (blue), 5 (green), 10 (red), 20 (yellow) and 40 (purple) mini-batch updates per environment step. **Right**: (c,d) Mean episode return as a function of number of transitions seen (in millions) of ADPG-R (16 workers) on the Grasp (c) and StackInHand (d) task. Same colors as in (a,b).

**Asynchronous DPG**   Increasing the number of update steps relative to the number of environment interactions greatly improves the data efficiency but also dramatically increases compute time. When the overall run time is dominated by the network updates it may scale linearly with the number of replay steps. In this setting experiments can quickly become impractical and parallelizing computation can provide a solution. Similarly, in a robotics setup the overall run time is typically dominated by the collection of interactions. In this case it is desirable to be able to collect experience from multiple robots simultaneously (e.g. as in (Yahya et al., 2016; Gu et al., 2016a)).

We therefore develop an asynchronous version of DPG that allows parallelization of training and environment interaction by combining multiple instances of an DPG-R actor and critic that each share their network parameters and can be configured to either share or have independent experience replay buffers. This is inspired by the A3C algorithm proposed in (Mnih et al., 2016), and also analogous to (Gu et al., 2016a; Yahya et al., 2016): We employ asynchronous updates whereby each worker has its own copy of the parameters and uses it for computing gradients which are then applied to a shared parameter instance without any synchronization. We use the Adam optimizer (Kingma & Ba, 2014) with local non-shared first-order statistics and a single shared instance of second-order statistics. The pseudo code of the asynchronous DPG-R is shown in algorithm box 1.

---

**Algorithm 1** (A)DPG-R algorithm

---

Initialize global shared critic and actor network parameters:
$\theta^{Q''}$ and $\theta^{\mu''}$
**Pseudo code for each learner thread:**
Initialize critic network $Q(s, a|\theta^Q)$ and policy network $\mu(s|\theta^\mu)$ with weights $\theta^Q$ and $\theta^\mu$.
Initialize target network $Q'$ and $\mu'$ with weights: $\theta^{Q'} \leftarrow \theta^Q, \theta^{\mu'} \leftarrow \theta^\mu$
Initialize replay buffer $R$
**for** episode = 1, M **do**
   Receive initial observation state $s_1$
   **for** t = 1, T **do**
      Select action $a_t = \mu(s_t|\theta^\mu) + \mathcal{N}_t$ according to the current policy and exploration noise
      Perform action $a_t$, observe reward $r_t$ and new state $s_{t+1}$
      Store transition $(s_t, a_t, r_t, s_{t+1})$ in $R$
      **for** update = 1, R **do**
         Sample a random minibatch of $N$ transitions $(s_i, a_i, r_i, s_{i+1})$ from $R$
         Set $y_i = r_i + \gamma Q'(s_{i+1}, \mu'(s_{i+1}|\theta^{\mu'})|\theta^{Q'})$
         Perform asynchronous update of the shared critic parameters by minimizing the loss:
         $L = \frac{1}{N} \sum_i (y_i - Q(s_i, a_i|\theta^Q)^2)$
         Perform asynchronous update of the shared policy parameters using the sampled gradient:

$$\nabla_{\theta^{\mu''}} \mu|_{s_i} \approx \frac{1}{N} \sum_i \nabla_a Q(s, a|\theta^Q)|\nabla_{\theta^\mu} \mu(s|\theta^\mu)|_{s_i}$$

         Copy the shared parameters to the local ones: $\theta^Q \leftarrow \theta^{Q''}, \theta^\mu \leftarrow \theta^{\mu''}$
         Every S update steps, update the target networks: $\theta^{Q'} \leftarrow \theta^Q, \theta^{\mu'} \leftarrow \theta^\mu$
      **end for**
   **end for**
**end for**

---

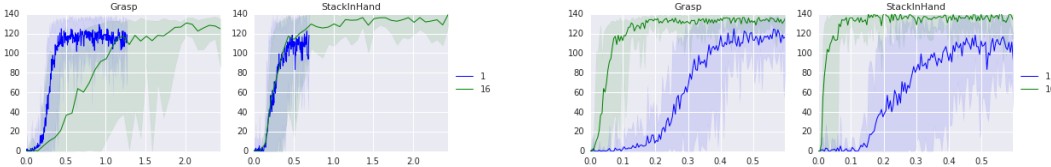

Figure 3: Data-efficiency and computational efficiency of ADPG-R. **Left**: Performance of 16 workers vs single worker in terms of environment transitions (x-axis is millions of transitions; total for all workers) for *Grasp* and *StackInHand* tasks. **Right**: Performance as a function of "wallclock" time (per-worker). Both with best replay step and learning rate selection.

Figure 2 (right) compares the performance of ADPG-R for different number of update steps and 16 workers (all workers performing both data collection and computing updates). Similar to Fig. 2 (left) we find that increasing the ratio of update steps per environment steps improves data efficiency, although the effect appears to be somewhat less pronounced than for DPG-R.

Figure 3 (left) directly compares the single-worker and asynchronous version of DPG-R. In both cases we choose the best performing number of replay steps and learning rate. As we can see, the use of multiple workers does not affect overall data efficiency for StackInHand but it reduced roughly in half for Grasp, with the note that the single worker still hasn't quite converged.

Figure 3 (right) plots the same data but as a function of environment steps *per worker*. This measure corresponds to the optimal wall clock efficiency that we can achieve, under the assumption that communication time between workers is negligible compared to environment interaction and gradient computation (this usually holds up to a certain degree of parallelization). The theoretical wall clock time for 16 workers is about 16x lower for StackInHand and roughly 8x lower for Grasp.

Overall these results show that distributing neural network training and data collection across multiple computers and robots can be an extremely effective way of reducing the overall run time of experiments and thus making it feasible to run more challenging experiments. We make extensive use of asynchronous DPG for remaining the experiments.

## 6    COMPOSITE SHAPING REWARDS

The reward function in the previous section was "sparse" or "pure" reward where a reward of 1 was given for states that correspond to successful task completion (brick lifted above 3cm for grasp; for stack) and 0 otherwise. For this reward to be useful it is necessary that the agent enters the goal region at least some of the time. While possible for each of the two subtasks in isolation, this is highly unlikely for the full task: without further guidance naïve random exploration is very unlikely to lead to a successful grasp-and -stack as we experimentally verify in Figure 4.

One solution are informative shaping rewards that provide a learning signal even for simple exploration strategies, e.g. by embedding information about the value function in the reward function.

This is a convenient way of embedding prior knowledge about the solution and is a widely and successfully used approach for simple problems. For complex sequential or compositional tasks such as the one we are interested in here, however, a suitable reward function is often non-obvious and may require considerable effort and experimentation. In this section we propose and analyze several reward functions for the full Stack task, and provide a general recipe that can be applied to other tasks with compositional structure.

Shaping rewards are often defined using a distance from or progress towards a goal state. Analogously our composite (shaping) reward functions return an increasing reward as the agent completes *components* of the full task. They are either piece-wise constant or smoothly varying across different regions of the state space that correspond to completed subtasks. In the case of Stack we use the following reward components (see the Appendix):

| Sparse reward components | | |
|---|---|---|
| Subtask | Description | Reward |
| Reach Brick 1 | hypothetical pinch site position of the fingers is in a box around the first brick position | 0.125 |
| Grasp Brick 1 | the first brick is located at least 3cm above the table surface, which is only possible if the arm is holding the brick | 0.25 |
| Stack Brick 1 | bricks stacked | 1.00 |
| **Smoothly varying reward components** | | |
| Reaching to brick 1 | distance of the pinch site to the first brick - non-linear bounded | [0, 0.125] |
| Reaching to stack | while grasped: distance of the first brick to the stacking site of the second brick - non-linear bounded | [0.25, 0.5] |

These reward components can be combined in different ways. We consider three different composite rewards in additional to the original sparse task reward:

**Grasp shaping**: *Grasp brick 1* and *Stack brick 1*, i.e. the agent receives a reward of 0.25 when brick 1 has been grasped and a reward of 1.0 after completion of the full task.

**Reach and grasp shaping**: *Reach brick 1*, *Grasp brick 1* and *Stack brick 1*, i.e. the agent receives a reward of 0.125 when close to brick 1, a reward of 0.25 when brick 1 has been grasped, and a reward of 1.0 after completion of the full task.

**Full composite shaping**: the sparse reward components as before in combination with the distance-based smoothly varying components

A full description of the reward functions is provided in the Appendix.

Figure 4 shows the results of learning with the above reward functions (blue traces). No progress on the full task is made when learning with the sparse reward only. *Grasp shaping* allows the agent to learn to grasp but learning is very slow. *Reach and grasp shaping* substantially reduces the time to successful grasping but learning does not progress beyond. Only with *Full composite shaping*, i.e. with an additional intermediate reward component as in continuous reach, grasp, stack is the full stacking task solved.

The actual reward functions given above are specific to the stacking task. But the general principle, a piecewise-constant sequence of rewards that increases as components of the tasks are completed, augmented with simple smoothly varying rewards that guide towards completion of individual sub-tasks should be widely applicable. It is important to note that the above reward functions do not describe all aspects of the task solution: we do not tell the agent how to grasp or stack but merely to bring the arm into a position where grasping (stacking) can be discovered from exploration and the sparse reward component. This eases the burden on the designer and is less likely to change the optimal solution in unwanted ways.

## 7    LEARNING FROM INSTRUCTIVE STATES

In the previous section we described a strategy for designing effective compositional reward functions that alleviate the burden of exploration. However, designing such rewards can still be error prone and we did indeed encounter several unexpected failure cases as shown in the supplemental video (https://www.youtube.com/watch?v=7vmXOGwLq24) and detailed in the Appendix. Furthermore, suitable rewards may rely on privileged information not easily available in a real robotics setup. In this section we describe a second, complementary strategy for embedding prior knowledge into the training process and improving exploration.

Specifically we propose to let the distribution of states at which the learning agent is initialized at the beginning of an episode reflect the compositional nature of the task: e.g., instead of initializing the agent at the beginning of the full task with both bricks on the table, we can initialize the agent occasionally with the brick already in its hand and thus prepared for stacking in the same way as when learning the subtask *StackInHand* in section 5.

More generally, we can initialize episodes with states taken from anywhere along or close to successful trajectories. Suitable states can be either manually defined (as in section 5), or they can be obtained from a human demonstrator or a previously trained agent that can partially solve the task. This can be seen as a form of apprenticeship learning in which we provide teacher information by influencing the state visitation distribution. Unlike many other forms of imitation or apprenticeship learning, however, this approach requires neither complete trajectories nor demonstrator actions.

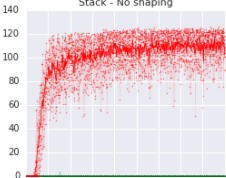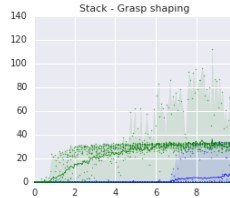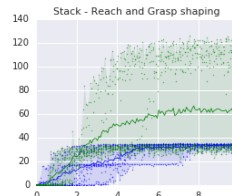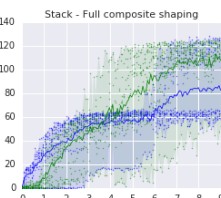

Figure 4: Effect of different reward shaping strategies and starting state distributions for the composite *Stack* task. Left to right: (a) No reward shaping; (b,c,d) reward shaping as explained in main text. Colors indicate starting states: Both bricks on the table (blue); manually defined initial states (green); and initial states continuously on solution trajectories (red). On all plots, x-axis is millions of transitions of total experience and y-axis is mean episode return. Policies with mean return over 100 robustly perform the full Stack from different starting states. Without reward shaping and basic start states only (a, blue) there is no learning progress. Instructive start states allow learning even with very uninformative sparse rewards indicating only overall task success (a,red).

We perform experiments with two methods for generating the starting states. The first one uses the manually defined initial states from section 5 (both bricks located on the table or in states where the first brick is already in the gripper as if the agent just performed a successful grasp). The second method initializes the learning agent at start states sampled randomly from successful demonstration trajectories (derived from agents previously trained end-to-end on the compositional reward).

The results of these experiments are shown in Figure 4. Green traces show results for the four reward functions from section 6 in combination with the manually defined start states (from section 5). While there is still no learning for the sparse reward case, results obtained with all other reward functions are improved. In particular, even for the second simplest reward function (*Grasp shaping*) we obtain some controllers that can solve the full task. Learning with the full composite shaping reward is faster and more robust than without the use of instructive states.

The leftmost plot of Figure 4 (red trace) shows results for the case where the episode is initialized anywhere along trajectories from a pre-trained controller (which was obtained using full composite shaping; rightmost blue curve). We use this start state distribution in combination with the basic sparse reward for the overall case (*Stack without shaping*). Episodes were configured to be 50 steps, which we found to be better suited to this setup with assisted exploration. During testing we still used episodes with 150 steps as before (so that the traces are comparable). We can see a large improvement in performance in comparison to the two-state method variant even in the absence of any shaping rewards. We can learn a robust policy for all seeds within a total of 1 million environment transitions — less than 1 hour of interaction time on 16 simulated robots.

These results suggest that an appropriate start state distribution not only speeds up learning, it also allows simpler reward functions to be used. In our final experiment we found that the simplest reward function (i.e. only indicating overall experimental success) was sufficient to solve the task. In this case the robustness of trained policies to starting state variation is also encouraging. Over 1000 test trials we obtain 99.2% success for *Grasp*, 98.2% for *StackInHand*, and 95.5% for the full *Stack* task.

## 8 CONCLUSION

We have introduced two extensions to the DDPG algorithm which make it a practical method for learning robust policies for complex continuous control tasks. We have shown that by decoupling the frequency of network updates from the environment interaction we can dramatically improve data-efficiency. Parallelizing data acquisition and learning substantially reduces wall clock time. In addition, we presented two methods that help to guide the learning process towards good solutions and thus reduce the pressure on exploration strategies and speed up learning. In combination these contributions allow us to solve a challenging manipulation problem end-to-end, suggesting that many hard control problems lie within the reach of modern learning methods.

It is of course challenging to judge the transfer of results in simulation to the real world. We have taken care to design a physically realistic simulation, and in initial experiments, which we have performed both in simulation and on the physical robot, we generally find a good correspondence

of performance and learning speed between simulation and real world. This makes us optimistic that performance numbers may also hold when going to the real world. A second limitation of our simulated setup is that it currently uses information about the state of the environment would require additional instrumentation of the experimental setup, e.g. to determine the position of the two bricks in the work space. These are issues that need to be addressed with care as experiments move to robotics hardware in the lab. Nevertheless, the algorithms and techniques presented here offer important guidance for the application of deep reinforcement learning methods to dexterous manipulation on a real robot.

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

# Appendix:

## 9   DDPG AND OTHER ALGORITHMS

DDPG bears a relation to several other recent model free RL algorithms: The NAF algorithm (Gu et al., 2016c) which has recently been applied to a real-world robotics problem (Gu et al., 2016a) can be viewed as a DDPG variant where the Q-function is quadratic in the action so that the optimal action can be easily recovered directly from the Q-function, making a separate representation of the policy unnecessary. DDPG and especially NAF are the continuous action counterparts of DQN (Mnih et al., 2015), a Q-learning algorithm that recently re-popularized the use of experience replay and target networks to stabilize learning with powerful function approximators such as neural networks. DDPG, NAF, and DQN all interleave mini-batch updates of the Q-function (and the policy for DDPG) with data collection via interaction with the environment. These mini-batch based updates set DDPG and DQN apart from the otherwise closely related NFQ and NFQCA algorithms for discrete and continuous actions respectively. NFQ (Riedmiller, 2005) and NFQCA (Hafner & Riedmiller, 2011) employ the same basic update as DDPG and DQN, however, they are batch algorithms that perform updates less frequently and fully re-fit the Q-function and the policy network after every episode with several hundred iterations of gradient descent with Rprop (Riedmiller & Braun, 1993) and using full-batch updates with the entire replay buffer. The aggressive training makes NFQCA data efficient, but the full batch updates can become impractical with large networks, large observation spaces, or when the number of training episodes is large. Finally, DPG can be seen as the deterministic limit of a particular instance of the stochastic value gradients (SVG) family (Heess et al., 2015), which also computes policy gradient via back-propagation of value gradients, but optimizes stochastic policies.

|  | Discrete | Continuous |
|---|---|---|
| Mini-batch learning | | |
| Target networks | DQN | DDPG, NAF |
| Full-batch learning with Rprop | | |
| Parameter resetting | NFQ | NFQCA |

## 10   REWARD FUNCTION

In this section we provide further details regarding the composite reward functions described in the main text. For our experiments we derived these from the state vector of the simulation, but they could also be obtained through instrumentation in hardware. The reward functions are defined in terms of the following quantities:

- $b_z^{(1)}$: height of brick 1 above table
- $s_{\{x,y,z\}}^{B1}$: x,y,z positions of site located roughly in the center of brick 1
- $s_{\{x,y,z\}}^{B2}$: x,y,z positions of site located just above brick 2, at the position where $s^{B1}$ will be located when brick 1 is stacked on top of brick 2.
- $s_{\{x,y,z\}}^{P}$: x,y,z positions of the pinch site of the hand – roughly the position where the fingertips would meet if the fingers are closed..

### 10.1   SPARSE REWARD COMPONENTS

Using the above we can define the following conditions for the successful completion of subtasks:

**Reach Brick 1**   The pinch site of the fingers is within a virtual box around the first brick position.

$$\text{reach} = (|s_x^{B1} - s_x^P| < \Delta_x^{\text{reach}}) \wedge (|s_y^{B1} - s_y^P| < \Delta_y^{\text{reach}}) \wedge (|s_z^{B1} - s_z^P| < \Delta_z^{\text{reach}}),$$

where $\Delta_{\{x,y,z\}}^{\text{reach}}$ denote the half-lengths of the sides of the virtual box for reaching.

**Grasp Brick 1**    Brick 1 is located above the table surface by a threshold, $\theta$, that is possible only if the arm is the brick has been lifted.

$$\text{grasp} = b_z^{(1)} > \theta$$

**Stack**    Brick 1 is stacked on brick 2. This is expressed as a box constraint on the displacement between brick 1 and brick 2 measured in the coordinate system of brick 2.

$$\text{stack} = (|C_x^{(2)}(s^{B1} - s^{B2})| < \Delta_x^{\text{stack}}) \wedge (|C_y^{(2)}(s^{B1} - s^{B2})| < \Delta_y^{\text{stack}}) \wedge (|C_z^{(2)}(s^{B1} - s^{B2})| < \Delta_z^{\text{stack}}),$$

where $\Delta_{\{x,y,z\}}^{\text{stack}}$ denote the half-lengths of the sides of the virtual box for stacking, and $C^{(2)}$ is the rotation matrix that projects a vector into the coordinate system of brick 2. This projection into the coordinate system of brick 2 is necessary since brick 2 is allowed to move freely. It ensures that the box constraint is considered relative to the pose of brick 2. While this criterion for a successful stack is quite complicated to express in terms of sites, it could be easily implemented in hardware e.g. via a contact sensor attached to brick 2.

## 10.2    SHAPING COMPONENTS

The full composite reward also includes two distance based shaping components that guide the hand to the brick 1 and then brick 1 to brick 2. These could be approximate and would be relatively simple to implement with a hardware visual system that can only roughly identify the centroid of an object. The shaping components of the reward are given as follows:

**Reaching to brick 1**    :

$$r_{S1}(s^{B1}, s^P) = 1 - \tanh^2(w_1 \| s^{B1} - s^P \|_2)$$

**Reaching to brick 2 for stacking**

$$r_{S2}(s^{B1}, s^{B2}) = 1 - \tanh^2(w_2 \| s^{B1} - s^{B2} \|_2).$$

## 10.3    FULL REWARD

Using the above components the reward functions we implement the composite reward functions described in the main text: *Stack*, *Grasp shaping*, *Reach and grasp shaping*, and *Full composite shaping* can be expressed as in equations (3, 4, 5, 6) below. These make use of the predicates above to determine whether which subtasks have been completed and return a reward accordingly.

$$r(b_z^{(1)}, s^P, s^{B1}, s^{B2}) = \begin{cases} 1 & \text{if stack}(b_z^{(1)}, s^P, s^{B1}, s^{B2}) \\ 0 & \text{otherwise} \end{cases} \tag{3}$$

$$r(b_z^{(1)}, s^P, s^{B1}, s^{B2}) = \begin{cases} 1 & \text{if stack}(b_z^{(1)}, s^P, s^{B1}, s^{B2}) \\ 0.25 & \text{if } \neg\text{stack}(b_z^{(1)}, s^P, s^{B1}, s^{B2}) \wedge \text{grasp}(b_z^{(1)}, s^P, s^{B1}, s^{B2}) \\ 0 & \text{otherwise} \end{cases} \tag{4}$$

$$r(b_z^{(1)}, s^P, s^{B1}, s^{B2}) = \begin{cases} 1 & \text{if stack}(b_z^{(1)}, s^P, s^{B1}, s^{B2}) \\ 0.25 & \text{if } \neg\text{stack}(b_z^{(1)}, s^P, s^{B1}, s^{B2}) \wedge \text{grasp}(b_z^{(1)}, s^P, s^{B1}, s^{B2}) \\ 0.125 & \text{if } \neg(\text{stack}(b_z^{(1)}, s^P, s^{B1}, s^{B2}) \vee \text{grasp}(b_z^{(1)}, s^P, s^{B1}, s^{B2})) \wedge \text{reach}(b_z^{(1)}, s^P, s^{B1}, s^{B2}) \\ 0 & \text{otherwise} \end{cases}$$

$$\tag{5}$$

$$r(b_z^{(1)}, s^P, s^{B1}, s^{B2}) = \begin{cases} 1 & \text{if stack}(b_z^{(1)}, s^P, s^{B1}, s^{B2}) \\ 0.25 + 0.25 r_{S2}(s^{B1}, s^P) & \text{if } \neg\text{stack}(b_z^{(1)}, s^P, s^{B1}, s^{B2}) \wedge \text{grasp}(b_z^{(1)}, s^P, s^{B1}, s^{B2}) \\ 0.125 & \text{if } \neg(\text{stack}(b_z^{(1)}, s^P, s^{B1}, s^{B2}) \vee \text{grasp}(b_z^{(1)}, s^P, s^{B1}, s^{B2})) \\ & \wedge \text{reach}(b_z^{(1)}, s^P, s^{B1}, s^{B2}) \\ 0 + 0.125 r_{S1}(s^{B1}, s^P) & \text{otherwise} \end{cases}$$

$$\tag{6}$$

## 10.4 COMPOSITE SHAPING REWARDS: FAILURE CASES

We encountered several unexpected failure cases while designing the reward function components. Examples of these are shown in the supplemental video (https://www.youtube.com/watch?v=7vmXOGwLq24). (1) The choice of reach and grasp reward components led to a grasp from which successful stacking was then no longer possible. (2) The agent learned to grasp the first brick and reach to the second but would then not stack because, due to an inappropriate height threshold the grasp reward would cease to be provided before the stack reward would be received. (3) The agent learns to flip the brick rather than lift it off the table because it receives a grasping reward calculated with an inappropriate reference point on the brick (which could be brought above the height threshold by merely bringing the brick on its side).

