# OpenReview forum: "Data-efficient Deep Reinforcement Learning for Dexterous Manipulation"
_ICLR.cc/2018/Conference — Reject_

### Official Review · AnonReviewer1 · 2017-11-24
**Neither very innovative nor very strong evaluations**

**Rating:** 4
**Confidence:** 4

**Review:**

I already reviewed this paper for R:SS 2017. There were no significant updates in this version, see my largely identical detailed comment in "Official Comment"

Quality
======
The proposed approaches make sense but it is unclear how task specific they are.

Clarity
=====
The paper reads well. The authors cram 4 ideas into one paper which comes at the cost of clarity of each of them.

Originality
=========
The ideas on their own are rather incremental.

Significance
==========
It is unclear how widely applicable the ideas (and there combination) are an whether they would transfer to a real robot experiment. As pointed out above the ideas are not really groundbreaking on their own.

Pros and Cons (from the RSS AC which sums up my thoughts nicely)
============
+ The paper presents and evaluates a collection of approaches to speed learning of policies for manipulation tasks.
+ Improving the data efficiency of learning algorithms and enabling learning across multiple robots is important for practical use in robot manipulation.
+ The multi-stage structure of manipulation is nicely exploited in reward shaping and distribution of starting states for training.

- The techniques of asynchronous update and multiple replay steps may have limited novelty, building closely on previous work and applying it to this new problem.
- The contribution on reward shaping would benefit from a more detailed description and investigation.
- There is concern that results may be specific to the chosen task.
- Experiments using real robots are needed for practical evaluation.

---

### Official Review · AnonReviewer3 · 2017-11-25
**The contributions already appear in prior work**

**Rating:** 2
**Confidence:** 5

**Review:**

The authors propose to learn to pick up a block and put it on another block using DDPG. A few tricks are described, which I believe already appear in prior work. The discussion of results presented in prior work also has a number of issues. The claim of "data efficient" learning is not really accurate, since even with demonstrations, the method requires substantially more experience than prior methods. Overall, it's hard to discern a clear contribution, either experimentally or conceptually, and the excessive claims in the paper are very off-putting. This would perhaps make a reasonable robotics paper if it had a real-world evaluation and if the claims were scoped more realistically, but as-is, I don't think this work is ready for publication.

More detailed comments:

The two main contributions -- parallel training and asynchrony -- already appear in the Gu et al. paper. In fact, that paper demonstrates learning entirely in the real world, and substantially more efficiently than described in this paper. The authors don't discuss this at all, except a passing mention of Gu et al.

The title is not appropriate for this paper. The method is data-efficient compared to what? The results don't look very data efficient: the reported result is something on the order of 160 robot-hours, and 16 robot-hours with demonstration. That's actually dramatically less efficient than prior methods.

"our results on data efficiency hint that it may soon be feasible to train successful stacking policies by collecting interactions on real robots": Prior work already shows successful stacking policies on real robots, as well as successful pick-and-place policies and a variety of other skills. The funny thing is that many of these papers are actually cited by the authors, but they simply pretend that those works don't exist when discussing the results.

"We assess the feasibility of performing analogous experiments on real robotics hardware": I assume this is a typo, but the paper does not actually contain any real robotics hardware experiments.

"To our knowledge our results provide the first demonstration of end-to-end learning for a complex manipulation problem involving multiple freely moving objects": This was demonstrated by Finn et al. in "Deep Spatial Autoencoders for Visuomotor Learning," with training times that are a tiny fraction of those reported in this paper, and using raw images and real hardware.

"both rely on access to a well defined and fully observed state space": This is not true of the Finn et al. paper mentioned above.

---

### Official Review · AnonReviewer2 · 2017-11-27
**Multiple minor extensions**

**Rating:** 3
**Confidence:** 4

**Review:**

The title is too generic and even a bit misleading. Dexterous manipulation usually refers to more complex skills, like in-hand manipulation or using the fingers to turn an object, and not simple pick and place tasks. Reinforcement learning methods are generally aiming to be data-efficient, and the method does not seem designed specifically for dexterous manipulation (which is actually a positive point, as it is more general).

The paper presents two extensions for DDPG: multiple network updates per physical interactions, and asynchronous updates from multiple robots. As the authors themselves state, these contributions are fairly straightforward, and the contributions are largely based on prior works. The  authors do evaluate the methods with different parameter settings to see the effects on learning performance.

The simulation environment is fairly basic and seems unrealistic. The hand always starts close to the blocks, which are close together, so the inverse kinematics will be close to linear. The blocks are always oriented in the same direction and they can connect easily with no need to squeeze or wiggle them together. The task seems more difficult from the description in the paper, and the authors should describe the environment in more detail.

Does the robot learn to flip the blocks over such that they can be stacked? The videos show the
blocks turning over accidentally, but then the robot seems to give up. Having the robot learn to turn the blocks  would make for a more challenging task and a better policy.

The paper’s third contribution is a recipe for constructing shaped reward functions for composite tasks. The method relies on a predefined task structure (reach-grasp-stack) and is very similar to reward shaping already used in many other reinforcement learning for manipulation papers. A comparison of different methods for defining the rewards and a more formal description of the reward generation procedure would improve the impact of this section.  The authors should also consider using tasks with longer sequences of actions, e.g., stacking four blocks.

The fourth and final listed contribution is learning from demonstrated states. Providing the robot with prior knowledge and easier partial tasks will result in faster learning. This result is not surprising. It is not clear though how applicable this approach is for a real robot system. It effectively assumes that the robot can grasp the block and pick it up, such that it can learn the stacking part, while simultaneously still learning how to grasp the block and pick it up. For testing the real robot applicability, the authors should try having the robot learn the task without simulation resets.

What are the actual benefits of using deep learning in this scenario? The authors mention skill representations, such as dynamic motor primitives, which employ significantly more prior knowledge than a deep network. However, as demonstrations of the task are provided, the task is divided into steps, the locations of the objects and finger tips are given, a suitable reward function is provided, and the generalization is only over the object positions, why not train a set of DMPs and optimize them with some additional reinforcement learning? The authors should consider adding a Cartesian DMP policy as a benchmark, as well as discussing the benefits of the proposed approach given the prior knowledge.

---

### Comment · AnonReviewer1 · 2017-11-24
**Many interesting ideas, but neither very innovative nor realisitc**

The paper proposes 4 approaches to speed up deep RL: multiple replay steps, asynchronous updates, reward shaping, and selecting starting states. The various combinations of approaches are evaluated on a combined grasping and stacking task.
I agree with the ideas related to reward shaping and selecting starting states in general. The other two approaches are rather specific to deep RL but well justified there.
The paper is a nice illustration that completely uninformative rewards do not work for complex, sequential tasks but that we also have to be very careful with rewards that they do not lead to the agent exploiting the reward definition in undesirable ways.

Both a strength and a weakness is that the methods proposes a whopping 4 approaches to speed up learning. It comes across as somewhat incremental for each of them and does not allow the authors to go very much in depth.
My main points of criticism:
- The first 2 contributions are heavily based on prior work (as pointed out by the authors in Sect. 5, but not in the intro and summary). It is not really clear what is novel and what is just showing "works for this type of problems as well".
- The contribution on reward shaping is very interesting, but then the "meat" is described in 10 lines which are hard to follow and to generalize to new problems.
- The contribution on learning from starting states ends up using a pre-trained policy or demonstration. It is not really clear how much this still is RL and how much this is supervised learning. I.e., We could also pre-train the value fct. and policy based on the demonstrations in a supervised fashion. Then it would be interesting to evaluate the performance and see how much RL can additionally improve upon that.
- I am not convinces this can really be transferred to a real robot as claimed by the authors and that the ideas are really widely applicable

Quite a bit of the discussion on RL, robotics, and rendering this combination tractable could be shortened e.g. by referring to Kober, J; Bagnell, D.; Peters, J. (2013). Reinforcement Learning in Robotics: A Survey, International Journal of Robotics Research, 32, 11, pp.1238-1274.

other interesting reference on parallel updates etc.
W. Caarls and E. Schuitema, “Parallel Online Temporal Diﬀerence Learning for Motor Control,” IEEE Transactions on Neural Networks and Learning Systems, vol. 27, pp. 1–12, Jul 2015.

- If you have such a clear task decomposition (and make use of it for the reward shaping) why not learn the parts separately, e.g., in a hierarchical RL setting?

Minor comments
==============
- Introduction: you keep talking about end-to-end but never mention what the inputs and outputs are until much later (end-to-end is typically vision+proprioception to torque, here the position and orientation of the blocks are given).
- Section 4: what is the definition of "physically plausible simulation"?
- Section 4: The observation vector is not entirely clear: 9 DoFs of the robot (position + velocity) is clear. The observations of the blocks not so much: position (3 dim?) + orientation (3 dim?), but then no velocities? And what dimension are the relative distances? Full 6 dim per block? Scalar?
- Section 5 (multiple- mini batches):  so you make this a lot more aggressive, does re-using the samples so often not lead to overfitting?
- Section 5 (asynchronous DPG): Any explanation why the speed-up of Grasp (8x) is significantly lower than for StackInHand (16x) for the 16 workers?
- It would be nice to have axes label in the plots
- "hasn't" => "has not"

---

### Decision · Program_Chairs · 2018-01-29
**ICLR 2018 Conference Acceptance Decision**

**Decision:**

Reject

**Comment:**

The reviewers were quite unanimous in their assessment of this paper.

PROS:
1. The paper is relatively clear and the approach makes sense
2. The paper presents and evaluates a collection of approaches to speed learning of policies for manipulation tasks.
3. Improving the data efficiency of learning algorithms and enabling learning across multiple robots is important for practical use in robot manipulation.
4. The multi-stage structure of manipulation is nicely exploited in reward shaping and distribution of starting states for training.

CONS
1. Lack of novelty e.g. wrt to Finn et al. in "Deep Spatial Autoencoders for Visuomotor Learning"
2. The techniques of asynchronous update and multiple replay steps may have limited novelty, building closely on previous work and applying it to this new problem.
3. The contribution on reward shaping would benefit from a more detailed description and investigation.
4. There is concern that results may be specific to the chosen task.
5. Experiments using real robots are needed for practical evaluation.